**Data Availability Statement:** The data used to support the findings of this study are included in the article and its supporting information files.

# The practice of early mother-newborn skin-to-skin contact after delivery of healthy term neonate and associated factors among health care professionals at health facilities of Southwestern Oromia, Ethiopia: A cross-sectional study

**Dejene Edosa Dirirsa**[1]*, **Mukemil Awol Salo**[1], **Tinsae Abeya Geleta**[2], **Berhanu Senbata Deriba**[2], **Girma Tufa Melese**[3]

1 Department of Midwifery, College of Medicine and Health Sciences, Salale University, Fiche, Ethiopia,
2 Department of Public Health, College of Medicine and Health Sciences, Salale University, Fiche, Ethiopia,
3 Department of Midwifery, College of Medicine and Health Sciences, Bule Hora University, Bule Hora, Ethiopia

☉ These authors contributed equally to this work.
* dejenedosa43@gmail.com

## Abstract

### Introduction

Skin-to-skin contact between a mother and her new-born baby after birth is beneficial for both the mother and her baby. Although mother-newborn skin-to-skin contact after birth is an essential practice, it is limited to a small proportion of premature babies in low-income countries including Ethiopia. The aim of this study was to assess the practice of early mother-new-born skin-to-skin contact after the delivery of healthy term neonates and associated factors among health care professionals in Southwestern Oromia, Ethiopia.

### Methods

An institutional-based cross-sectional study was conducted to assess the practice of 286 health care practitioners towards early mother-new-born skin-to-skin contact after delivery. Data was collected using a pre-tested observational checklist and a self-administered questionnaire from March to April 2017. Epi Info 3.5 was used for data entry, while SPSS version 20 was used for cleaning and analyzing the data. To determine the association between outcome variable and independent variables, bivariate and multivariable logistic regressions were used with a 95% confidence interval and P <0.05. Frequency tables and charts were used to present the findings.

### Results

Only 128 (44.8%) of the study participants practiced mother-newborn skin-to-skin contact within the first hour of life after birth. Mother newborn skin-to-skin contact after birth was

**Funding:** The author(s) received no specific funding for this work.

**Competing interests:** The authors have declared that no competing interests exist.

**Abbreviations:** AOR, Adjusted odds ratio; BF, Breast feeding; CI, Confidence interval; COR, Crude odds ratio; KMC, Kangaroo mother care; NICU, Neonatal intensive care unit; PPH, Post-partum hemorrhage; SSC, Skin to skin contact; SPSS, Statistical Package for Social Science; WHO, World Health Organization.

found to be significantly associated with health professional's knowledge (AOR = 4, 95% CI = 1.7, 10), training (AOR = 7, 95% CI = 2.2, 21), complicated delivery (AOR = 0.12, 95% CI = 0.04, 0.4), and maternal chronic illness (AOR = 0.13, 95% CI = 0.03, 0.6).

## Conclusion

In general, the practice of health care providers on mother-newborn skin-to-skin contact in the first one hour after birth was low. Knowledge, training, childbirth related maternal complication, and maternal chronic illness were significant factors associated with the practice of mother-newborn skin-to-skin contact immediately after birth. Policies should be revised and enforced, with monitoring and awareness building through training among health care workers, to improve the practice of skin-to-skin contact between mothers and newborns shortly after birth.

## Introduction

### Background

Early skin-to-skin contact (SSC) between mother and newborn is defined as placing the bare baby's body on the mother's bare chest and covering the rest of the baby's body parts with a warm blanket within the first hour after birth. The first hour after delivery is an essential time for starting a newborn's feeding habits, such as searching for breast and sucking [1].

The supported suggestions indicated that the practice of skin-to-skin contact after birth has strong benefits for both mother and newborn, include: shortening of delivery of the placenta, decrease hemorrhage after birth by increasing the mother's oxytocin during the first hour, and decreases bad consequences of the 'stress of being born' of the newborn, thermoregulation, and promote breastfeeding [2].

The first day of life for the newborn is the most essential moment of survival. During this initial phase, newborns may need support to set up regular breathing and maintain normal body temperature and blood sugar levels to avoid potentially life-threatening situations [3].

Worldwide, 2.5 million child deaths occur in the first months of life in 2017 which accounts 47% of all child mortality. The majority of neonatal death (99%) occurs in low-income countries, and the important causes of death include prematurity, birth asphyxia, infection, birth defects, and hypothermia. A higher prevalence of neonatal hypothermia has been reported from countries with the highest neonatal mortality which can be preventable through the principle of SSC and initiating breastfeeding within the first hour after birth [4].

Even though the actual possibility of practicing SSC, currently only a few preterm babies in low & middle-income countries have access to these practices. The reason may be opposition among health care providers regarding providing SSC. This opposition could be due to fear of harm to the infant, lack of skill or training, time, institutional guidelines, and assistance to hand over the infant to the parent and/or monitor the infant's well-being [5, 6].

In medical training programs, the factors should be identified and ranked to emphasize the effectiveness of the implementation of SSC. Factors could affect the behavior of healthcare providers in implementing mother-infant skin-to-skin contact upon birth. For instance, inadequate equipment, insufficient human resources, and absence of structured programs for the proper implementation of SSC are factors associated with mother-newborn skin-to-skin contact [7].

Although the mother-newborn skin-to-skin contact is beneficial for both mother and newborn, in many developing-countries like Ethiopia, health care professionals separate mother

and newborn immediately after birth. In this case mother-newborn skin-to-skin contact is rarely used only to treat neonates who are clinically ill due to prematurity or other disorders. These babies are routinely separated from their mothers and placed in incubators or radiant warmers for long periods of time [3].

Ethiopia is one of the ten countries in the world with the highest rate of neonatal mortality, with an estimated 122,000 newborn mortality each year. Inadequate perinatal and postpartum care for the mother and newborn also exists. In Ethiopia, there is also a lack of evidence regarding mother-newborn skin-to-skin contact practices because several critical variables have not been identified recently by routine surveys such as the Demographic and Health Survey. In many hospitals across the country, mother-infant separation is common procedure [8].

The mother-infant SSC approach has emerged as the most effective neonatal care option at the hospital and community level. Consequently, both national and subnational research is becoming a foundation of the strategy to monitor progress at the desired level [9].

However, only a few types of studies on mother-infant skin-to-skin contact have been undertaken, and data on the proportion of SSC at the national level is lacking. As a result, the purpose of this study was to evaluate the practice of early mother-newborn skin-to-skin contact after the birth of healthy term neonates, as well as associated factors, among health care practitioners in Southwestern Ethiopian health institutions.

## Methodology

### Study design, area and period

Institutional-based cross-sectional study was carried out in Ilu Abba Bor and Bunno Beddelle districts of Oromia region south-western Ethiopia, from March to April 2017. Mettu and Bedelle, the capitals of both districts, are located in southwestern Ethiopia, 600 and 483 kilometers far from Addis Ababa, respectively. According to the 2007 Ethiopian national census the total population of the districts were over 1.2 million and 800,000 people, respectively. In the district, Mettu, Darimu, and Bedelle hospitals, as well as thirty-nine primary health care units were giving the delivery service during the study period.

### Populations

All health-care professionals in Ilu Abba Bor and Bunno Beddelle districts (Midwives, Nurses, Health officers, Emergency surgeons, General practitioners, Gynecologists, and Senior surgeons) who were giving delivery service at the health facilities were the study population.

### Eligibility criteria

The study included all health care practitioners working in the labor and delivery wards of health facilities in Ilu Abba Bor and Bunno Beddelle districts, with the exception of those who were seriously ill during data collection.

### Sampling technique & procedures

Using a convenient sampling technique, health care practitioners working in labor and delivery rooms at each health facility were selected as study participants during the data collection period. The number of individual participants was insufficient to justify using the probability sampling method. As a result, we decided to use all of the illegible participants who were interested to participate in our study. The study included 84 health care practitioners from three hospitals and 202 health care practitioners from a 39 health facilities in the districts.

## Data collection instruments and procedures

The data was gathered using a pre-tested observational checklist and a self-administered questionnaire that developed after review of several comparable works of literature and written in English [10–12]. The data was collected by eight BSc Midwives who were supervised by the principal investigator and had more experience in labor and delivery as well as knowledge of the skin-to-skin contact technique.

The practice of SSC was assessed using an observational checklist while clinicians were caring for newborns without their knowledge. The observation was carried out without the participants' awareness or knowledge of the specific care being observed, however consent was obtained for the assessment of general immediate new-born care. Following the observational assessment, all study participants were given a self-administered questionnaire based on their code, which was used to conduct a knowledge assessment. The questionnaire was validated by advisors, and additional professionals' comments were incorporated into the final instrument used for data collection.

## Data quality control

Data collectors were trained on subjects such as the content of the checklist and questionnaire, how to approach the study units, and the process how to assess practice of SSC and confidentiality before the actual data collection to assure the quality of the data. The checklist and questionnaire were pre-tested on 5% of the sample size at Agaro hospital one week before the actual data collection. Based on the results of the pre-test, unnecessary variables and ambiguous words were found and fixed.

## Study variables

**The dependent /outcome/ variable.**   Mother-newborn skin-to-skin contact practice status.

**The independent variables.**   Age, sex, marital status, educational level, work experience (in years), knowledge, training, and health of the mother, complications of delivery, mother's choice, and maternal request to put the newborn on her chest.

## Data analysis

Epi Info version 3.5 was used for data entry, while SPSS version 20 statistical software was used for data cleaning and analysis. The association between outcome variable and independent variables was determined using bivariate and multivariate logistic regression with a 95% confidence interval, and statistical significance was set at P 0.05. The result was presented by frequency, percentages, tables and chart.

## Operational and term definitions

**Skin-to-skin contact practice.**   Placing the bare baby's body on the mother's bare chest by covering the rest of the parts with a warm blanket within the first hour after birth.

**Good knowledge.**   Scoring ≥50% from knowledge measuring questionnaires.

**Poor knowledge.**   Scoring of <50% from knowledge measuring questionnaires.

**Practice.**   If the providers apply early skin-to-skin contact according to observational checklist used for this study which has 7 questions. (1. Was naked newborn put belly-down on his or her mother's bare abdomen or chest and then wrap both with warm cloth, with in the 1$^{st}$hr? 2. Was the newborn stays ≥1hr in skin to skin contact continuously? 3. Was the newborn allowed to contact with the nipple? 4. Was the newborn's head covered with warm cloth

or with cap? 5. Was the newborn wrapped with warm cloth and put on the mother's abdomen within the 1<sup>st</sup> hour? 6. Was the newborn wrapped with warm cloth and put at the side of the mother on the bed? 7. Was the newborn wrapped with warm cloth and/or stay more than an hour under the heater?) [12, 13].

### Ethical consideration

Ethio-Canada MCH project and Saint Paul Millennium Medical College (SPMMC) Research Ethical Review Committee gave the approval. Supportive letter was written to each health offices of Ilu Abba Bor and Bunno Beddelle districts.

Ethical considerations were taken into account at every stage of the study. The study participants were given a full description of the objective of the study, purpose, benefit, and importance. Data was collected after complete informed verbal consent was obtained, and the information was kept confidential throughout the process.

## Results

### Socio-demographic characteristics of health care professionals providing newborn care

During the study period, 286 health care providers were observed while providing care to new-born. One hundred eighty (62.9%) of the study participants were between the ages of 21 and 30, 168 (58.7%) of them were females and 160 (55.9%) of them being married. In terms of respondents' educational backgrounds, 158 (55.3%) of health care professionals had a BSc degree, while 184 (64.3%) had 1 to 5 years of experience. The majority of health care practitioners, 228 (79.7%), did not have the opportunity to participate in the mother-newborn skin-to-skin contact practice training *(Table 1)*.

**Knowledge of health care providers on mother-newborn skin-to-skin contact.** Skin-to-skin contact between a mother and her newborn is beneficial to effective breastfeeding, according to the majority of health care practitioners (82.5%). Two hundred twenty (76.9%) of the study participants correctly identified the appropriate time to begin mother-newborn skin-to-skin contact, whereas 170 (59.4%) had awareness on the option of practicing skin-to-skin contact while managing complicated labor and delivery *(Table 2)*.

**The practice of mother-newborn skin-to-skin contact.** From the total study participants, 128 (44.7%) health care providers placed a naked new-born on the mother's bare abdomen for at least 30 minutes after birth (within the first 1hr), allowing contact with the nipple, and wrapped the new-born with a warm cloth, while 52 (18.2%) study participants placed the new-born on the mother's abdomen after wrapping the new-born with cloth, and 36 (12.6%) health care providers wrapped the new-born with cloth and placed it at the side of the mother on the bed, whereas some study subjects 70 (24.5%) wrapped the new-born with a cloth and put under the heater.

### Reasons for not practicing mother-newborn skin-to-skin contact

In the absence of maternal complications and/or disease, 96 (33.6%) health care providers were not practicing mother-new-born SSC, and they asked for specific reasons that might preclude them from practicing mother-new-born SSC immediately after birth. The most frequently answered reason for not practicing mother-new-born SSC was, keeping newborn under the heater is a usual pattern of activity (54.2%), followed by a misunderstanding that says, putting the new-born on the mother's abdomen after wrapping it with warm cloth is the right procedure (18.7%) (Fig 1).

**Table 1. Socio-demographic characteristics of health care professionals providing newborn care during observation in labor and delivery wards of health institutions in the Ilu Abba Bor and Bunno Beddelle Zones, Oromia, Ethiopia, 2017.**

| Variables | | Number (n = 286) | Percentage (%) |
|---|---|---|---|
| Age in years | 21–30 | 180 | 62.9 |
| | 31–40 | 72 | 25.2 |
| | 41–50 | 34 | 11.9 |
| Sex | Male | 118 | 41.3 |
| | Female | 168 | 58.7 |
| Marital status | Never married | 126 | 44.1 |
| | Married*** | 160 | 55.9 |
| Educational level | Diploma | 128 | 44.7 |
| | BSc degree and above | 158 | 55.3 |
| Years of experience | 1–5 | 184 | 64.3 |
| | 6–10 | 26 | 9.1 |
| | 11–15 | 36 | 12.6 |
| | 16 and above | 40 | 14 |
| Training | Not trained | 228 | 79.7 |
| | Trained | 58 | 20.3 |

Married

*** = include married, separated, divorced and widowed.

## Factors associated with practice of mother-newborn skin to skin contact of health care providers giving new born care

Educational level, knowledge, maternal request for SSC, health care provider training, complicated delivery, and maternal illness were significantly associated with practice of mother-

**Table 2. Knowledge of health care providers on mother-newborn skin-to-skin contact of Ilu Abba Bor and Bunno Beddelle Zones health facilities, Oromia, Ethiopia 2017.**

| Variables | | Frequency (%) |
|---|---|---|
| The correct procedure for practice of mother-SSC | Put naked newborn on mother's bare abdomen and cover | 188(65.7) |
| | Wrap & put on the mother's abdomen without cloth | 98(34.3) |
| The appropriate time to start mother-newborn SSC | Within the first 1hour | 220(76.9) |
| | After hour | 66(23.1) |
| The minimum time duration for the newborn to stay on SSC | 30minutes | 202(70.6) |
| | 1 hour | 84 (29.4) |
| Practice of SSC is possible during managing complication aroused from delivery | Yes | 170(59.4) |
| | No | 116(40.6) |
| Skin to skin contact prevents neonatal hypothermia | Yes | 182(63.6) |
| | No | 104(36.4) |
| Skin to skin contact promotes effective breast feeding | Yes | 236(82.5) |
| | No | 50(17.5) |
| SSC improves neonatal breathing, prevents neonatal infection, accelerates involution and prevents PPH | Yes | 174(60.8) |
| | No | 112(39.2) |
| Over all knowledge | Good Knowledge | 196(68.5) |
| | Poor knowledge | 90(31.5) |

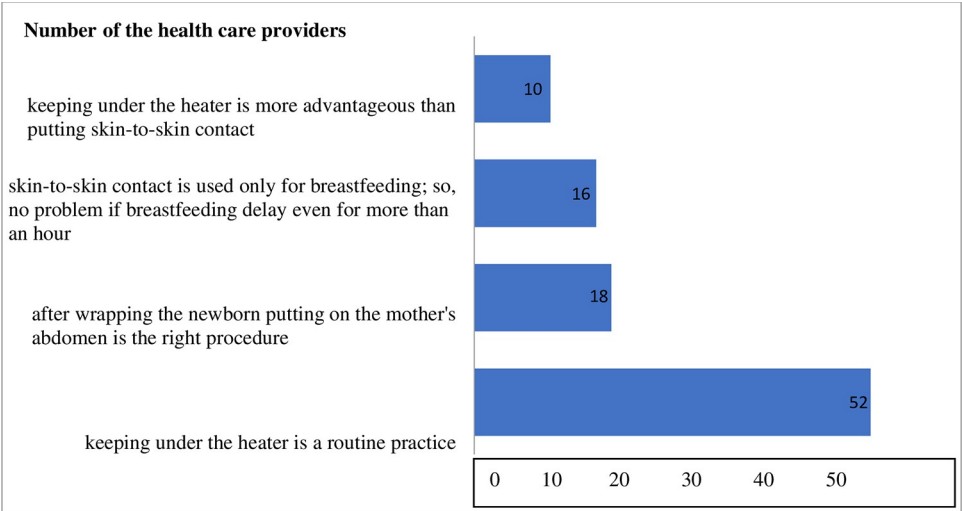

**Fig 1. Reasons for not practicing mother-new born skin-to-skin contact of health care providers giving new born care during observation in Ilu Abba Bor and Bunno Beddelle zones health facilities, Oromia, Ethiopia 2017.**

newborn skin-to-skin contact in bivariate logistic regression. Health care provider knowledge about SSC, training on SSC, complicated delivery, and maternal chronic illness were found to be significantly associated with mother-newborn skin-to-skin contact at a p-value less than 0.05 with a 95 percent confidence interval in a multivariable logistic regression.

The result of multivariable logistic regression indicated that health care professionals who had good knowledge were 4 times more likely to practice mother-newborn skin-to-skin contact than those who had poor knowledge (AOR = 4, 95% CI = 1.7, 10). Health care professionals who had immediate newborn care training were 7 times more likely to practice mother-newborn skin-to-skin contact than those who had no training (AOR = 7, 95% CI = 2, 21).

The study participants who faced complicated delivery were 88% less likely to practice mother-newborn SSC as compared to those who did not face the complicated delivery (AOR = 0.12, 95% CI = 0.03, 0.4), whereas the study subjects who attended delivery with maternal chronic illness are 87% less likely to practice mother-newborn skin to skin contact when compared with those who attend the delivery without maternal illness (AOR = 0.13, 95% CI = 0.03, 0.6) *(Table 3)*.

## Discussion

According to the findings of the current study, 44.7% of health care practitioners practiced skin-to-skin contact between the mother and the newborn within the first hour after birth. This finding is in line with the study conducted in Hawi district, Northwestern Ethiopia, which found that 47.8% of participants put the baby in skin-to-skin contact after removing the wet towel [8].

The finding is lower than two studies conducted in Tigray's Northwestern and Eastern district health facilities, which found that 52 percent and 86.4 percent of participants practiced skin-to-skin contact to prevent hypothermia [7]. This inconsistency could be due to the difference in study population, socio-demographic characteristics and socio-cultural of the societies.

Good knowledge of health the care health providers was one of the predictors of practice of mother newborn SSC. This is in line with findings from a study conducted in central Ethiopia, which indicated that a lack of good knowledge caused Midwives to ineffectively practice

**Table 3. Bivariate and multivariate logistic regression for factors associated with practice of skin-to-skin contact, in health care providers giving newborn care observation in Ilu Abba Bor and Bunno Beddelle Zones health facilities, Oromia, Ethiopia 2017.**

| Variables | Category | Practice | | COR | AOR |
|---|---|---|---|---|---|
| | | Yes | No | (95% CI) | (95% CI) |
| Educational level | Diploma | 44 (34.4%) | 84 (65.6%) | 1 | |
| | BSc degree+ | 84 (53.2%) | 74(46.8%) | 2.17(1.1, 4.3) | |
| Knowledge | Poor knowledge | 22 (24.4%) | 68 (75.6%) | 1 | |
| | Good Knowledge | 106 (54.1%) | 90 (45.9%) | .28 (.13, .6) | 4(1.7, 10)* |
| Maternal request | Not requested | 110 (41.7%) | 144 (58.3%) | 1 | |
| | Requested | 26 (81.25%) | 90 (45.9%) | 6.3(1.3, 30.3) | |
| Training | Not trained | 82 (36%) | 146 (64.0%) | 1 | |
| | Trained | 46 (79.3%) | 12 (20.7%) | 6.8(2.6, 18) | 7 (2, 21)* |
| Maternal complication | Not complicated | 118 (50%) | 118 (50%) | 1 | |
| | Complicated | 10 (20%) | 40 (80%) | .25 (.09, .7) | 0.12 (.03, .4)* |
| Maternal illness | Absent | 122 (48.8%) | 130 (51.2%) | 1 | |
| | Present | 4 (12.5%) | 28 (87.5%) | .15 (.03, .7) | 0.13(0.03, 0.6)* |

NB: 1 = Reference

* = p-value < 0.05.

mother-newborn skin-to-skin contact [9]. This is because the knowledge of health-care practitioners is required in order to practice the mother-newborn skin-to-skin contact.

The practice of mother-newborn SSC was found to be strongly associated to skin-to-skin contact or early newborn care training. The current finding is supported by research conducted in Jimma and Northwestern Tigray that found health care providers who took essential newborn care training were more likely to practice mother-newborn SSC than their counterparts [14]. This could be because of the study participants' skill and knowledge increased as a result of their training essential newborn care.

A complicated delivery and chronic maternal illness were also significantly associated to the practice of mother-new-born SSC right after birth. Study participants who attended a complicated delivery were 88% less likely than those who attended an uncomplicated delivery to practice mother-newborn SSC, and those who attended a delivery with chronic maternal illness were 87% less likely to practice mother-newborn SSC than those who attended a delivery without maternal illness. This is supported by research conducted at Georgetown University on provider utilization of maternal-infant SSC, which found that the majority of midwives used mother-newborn SSC in healthy mothers [10].

This could be because it is difficult for healthcare workers to practice mother-newborn skin-to-skin contact for mothers who have had a complicated birth or who have chronic maternal sickness. Some difficulties or illnesses may take longer to resolve, and the environment may not be favorable to skin-to-skin contact between the mother and the newborn.

## Conclusion

In general, health-care providers' practices regarding mother-new-born skin-to-skin contact within the first hour after birth was low. Knowledge status of health-care providers, training, complicated delivery and maternal chronic illnesses were factors significantly associated with mother-newborn SSC within the first 24 hours after birth.

Update and implement the policy on immediate newborn care, monitoring, and training health care providers on how to perform mother-newborn SSC right after birth is compulsory to implement the mother-newborn SSC.

## Supporting information

**S1 File. Questionaire—English version.**
(DOCX)

**S2 File. Questionaire—Local language version (Afaan Oromoo).**
(DOCX)

## Acknowledgments

All of my respects and thanks go to almighty "God," the provider of many benefits and wisdom, from the beginning. Second, I want to express my heartfelt gratitude to my friends for their unwavering support and encouragement during my efforts. I'd also like to thank the Ethio-Canada MCH project and Saint Paul Millennium Medical College (SPMMC) for providing me with the opportunity to work and for inspiring me to perform this research.

## Author Contributions

**Conceptualization:** Dejene Edosa Dirirsa, Tinsae Abeya Geleta.

**Data curation:** Dejene Edosa Dirirsa, Mukemil Awol Salo, Tinsae Abeya Geleta, Berhanu Senbata Deriba.

**Formal analysis:** Dejene Edosa Dirirsa, Mukemil Awol Salo, Tinsae Abeya Geleta, Berhanu Senbata Deriba, Girma Tufa Melese.

**Funding acquisition:** Dejene Edosa Dirirsa.

**Investigation:** Dejene Edosa Dirirsa, Berhanu Senbata Deriba.

**Methodology:** Dejene Edosa Dirirsa, Mukemil Awol Salo, Tinsae Abeya Geleta, Berhanu Senbata Deriba, Girma Tufa Melese.

**Project administration:** Dejene Edosa Dirirsa, Mukemil Awol Salo.

**Resources:** Dejene Edosa Dirirsa, Tinsae Abeya Geleta.

**Software:** Dejene Edosa Dirirsa, Mukemil Awol Salo, Berhanu Senbata Deriba, Girma Tufa Melese.

**Supervision:** Dejene Edosa Dirirsa, Mukemil Awol Salo, Tinsae Abeya Geleta, Berhanu Senbata Deriba.

**Validation:** Dejene Edosa Dirirsa, Berhanu Senbata Deriba, Girma Tufa Melese.

**Visualization:** Dejene Edosa Dirirsa, Tinsae Abeya Geleta, Berhanu Senbata Deriba, Girma Tufa Melese.

**Writing – original draft:** Dejene Edosa Dirirsa.

**Writing – review & editing:** Dejene Edosa Dirirsa, Girma Tufa Melese.

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
