## [Decision Letter · Decision Letter 0]

17 Aug 2021

PONE-D-21-11913

The practice of early mother-newborn skin-to-skin contact after delivery of healthy term neonate and associated factors among health care providers at health facilities of Southwestern Oromia, Ethiopia 2017.

PLOS ONE

Dear Dr. Dirirsa 

Thank you for submitting your manuscript to PLOS ONE. After careful consideration, we feel that it has merit but does not fully meet PLOS ONE’s publication criteria as it currently stands. Therefore, we invite you to submit a revised version of the manuscript that addresses the points raised during the review process. The paper needs major revision.

We look forward to receiving your revised manuscript.

Kind regards,

Prof Sajid Soofi

Academic Editor

PLOS ONE

1. Please ensure that your manuscript meets PLOS ONE's style requirements, including those for file naming. The PLOS ONE style templates can be found at https://journals.plos.org/plosone/s/file?id=wjVg/PLOSOne_formatting_sample_main_body.pdf and https://journals.plos.org/plosone/s/file?id=ba62/PLOSOne_formatting_sample_title_authors_affiliations.pdf.

2. Please amend your current ethics statement to address the following concerns: Please explain why written consent was not obtained, how you recorded/documented participant consent, and if the ethics committees/IRBs approved this consent procedure.

3. Please include additional information regarding the checklist and questionnaire used in the study and ensure that you have provided sufficient details that others could replicate the analyses. For instance, if you developed a questionnaire as part of this study and it is not under a copyright more restrictive than CC-BY, please include a copy, in both the original language and English, as Supporting Information. If the original language is written in non-Latin characters, for example Amharic, Chinese, or Korean, please use a file format that ensures these characters are visible.

4. Please state whether you validated the questionnaire prior to testing on study participants. Please provide details regarding the validation group within the methods section.

5. Thank you for stating the following in the Acknowledgments/Funding Section of your manuscript:

“The entire necessary costs (material and humanitarian) for the study were covered by Ethio-Canada MCH project funded this study.”

6. Please note that in order to use the direct billing option the corresponding author must be affiliated with the chosen institute. Please either amend your manuscript to change the affiliation or corresponding author, or email us at plosone@plos.org with a request to remove this option.

7. PLOS requires an ORCID iD for the corresponding author in Editorial Manager on papers submitted after December 6th, 2016. Please ensure that you have an ORCID iD and that it is validated in Editorial Manager. To do this, go to ‘Update my Information’ (in the upper left-hand corner of the main menu), and click on the Fetch/Validate link next to the ORCID field. This will take you to the ORCID site and allow you to create a new iD or authenticate a pre-existing iD in Editorial Manager. Please see the following video for instructions on linking an ORCID iD to your Editorial Manager account: https://www.youtube.com/watch?v=_xcclfuvtxQ

8. Please include your full ethics statement in the ‘Methods’ section of your manuscript file. In your statement, please include the full name of the IRB or ethics committee who approved or waived your study, as well as whether or not you obtained informed written or verbal consent. If consent was waived for your study, please include this information in your statement as well.

Additional Editor Comments (if provided):

This is an interesting study focusing on the practice of skin-to-skin contact of term neonates. Please review the comments and revise the paper.

Reviewers' comments:

Reviewer's Responses to Questions

**Comments to the Author**

1. Is the manuscript technically sound, and do the data support the conclusions?

Reviewer #1: Partly

Reviewer #2: Partly

2. Has the statistical analysis been performed appropriately and rigorously? 

Reviewer #1: Yes

Reviewer #2: Yes

3. Have the authors made all data underlying the findings in their manuscript fully available?

Reviewer #1: Yes

Reviewer #2: Yes

4. Is the manuscript presented in an intelligible fashion and written in standard English?

Reviewer #1: No

Reviewer #2: Yes

5. Review Comments to the Author

Reviewer #1: • How did you measure the practice? Relevant results should be stated

• Explain more about tools and Validity and reliability of the tools

• Remove results (For example (P=0.002, AOR=4, CI=1.7, 10…) from the discussion section

• You can use the following articles in the introduction and discussion sections

1. Karimi FZ, Miri HH, Khadivzadeh T, Maleki-Saghooni N. The effect of mother-infant skin-to-skin contact immediately after birth on exclusive breastfeeding: a systematic review and meta-analysis. Journal of the Turkish German Gynecological Association. 2020 Mar;21(1):46.

2. Karimi FZ, MIRI HH, Salehian M, Khadivzadeh T, Bakhshi M. The Effect of Mother-Infant Skin to Skin Contact after Birth on Third Stage of Labor: A Systematic Review and Meta-Analysis. Iranian journal of public health. 2019 Apr;48(4):612.

3. Karimi A, Bagheri S, Khadivzadeh T, Mirzaii Najmabadi Kh. The Effect of an Interventional Program, Based on the Theory of Ethology, on Breastfeeding Competence of Infants. Iranian Journal of Neonatology 2014; 5(3): 10-12.

4. Karimi A, khadivzadeh T, Bagheri S. Effect of immediate and continuous mother- infant skin to skin contact on breastfeeding selfefficacy of primiparous women. Women and birth 2014; 27:37-40.

5. Karimi FZ, Khadivzadeh T, Saeidi M, Bagheri S. The Effect of Kangaroo Mother Care Immediately after Delivery on Mother-infant Attachment and on Maternal Anxiety about the Baby 3- Months after Delivery: a Randomized Controlled Trial. Int J Pediatr 2016; 4(9): 3561-70

6. Karimi A, Tara F, Khadivzadeh T, Aghamohammadian Sharbaf HR. The Effect of Skin to Skin Contact Immediately after Delivery on the Maternal Attachment and Anxiety Regarding Infant. The Iranian Journal of Obstetrics, Gynecology and Infertility 2013; 16(67): 7-15.

7. Karimi FZ, Bagheri S, Tara F, Khadivzadeh T, Mousavi Bazaz SM. Effect of Kangaroo Mother Care on breastfeeding self-efficacy in primiparous women, 3 month after child birth. The Iranian Journal of Obstetrics, Gynecology and Infertility 2014; 17(120): 1-8.

8. Khadivzadeh T, Karimi FZ, Tara F, Bagheri S. The Effect of Postpartum Mother– Infant Skin-to-Skin Contact on Exclusive Breastfeeding In neonatal period: A Randomized Controlled Trial. Int J Pediatr 2016; 4(5): 5409-17.

9. Khadivzadeh, T., Karimi, F., Tara, F. Effects of early mother-neonate skin-to-skin contact on the duration of the third stage of labor: A randomized clinical trial. The Iranian Journal of Obstetrics, Gynecology and Infertility, 2018; 21(2): 23-29

10. Karimi FZ, Sadeghi R, Maleki-Saghooni N, Khadivzadeh T. The effect of mother-infant skin to skin contact on success and duration of first breastfeeding: A systematic review and meta-analysis. Taiwanese Journal of Obstetrics and Gynecology. 2019 Jan 1; 58(1):1-9.

Reviewer #2: General: This is an interesting study that focus on the practice of skin to skin contact post delivery among new mothers in Ethiopia.

Introduction: Although the authors mentioned about Ethiopia's maternal care. However, the authors should elaborate on the national policy and maternal care services. The world prevalence is not very important in this case; rather the reader would be more interested on Ethiopia's national policy on maternal delivery and care; as well as the maternity services / labour services available. Why are 90% of birth occuring at home? is that cultural practice or due to insufficient maternity hospital? transport or distance to hospital? Suggest to focus solely on Ethiopia, describing its maternal healthcare services, national policy, home birth rate and why it is so prevalent and then on the practices of skin-to-skin contact post delivery in Ethiopia; Also, to add if other similar studies have been done in Ethiopia and what are the current known findings (if any)

Statement of the problem: paragraph 5 - this paragraph is not clear in terms of what the authors are saying. Please reword the sentences.

Method: please explain why convenience sampling was applied? Also, the method states that pre-testing was done. What kind of pre-testing? was reliability of the questiionnaire measured? What was the result of the pre-testing? In data collection, it stated that the healthcare providers were unaware they were being observed however, consent was taken before they were recruited so, how was that achieved? Please explain.

Results: Please check table 3 maternal illness if your significance level is correct?

Discussion: could be better written at the end to tally all the four factors and how it can be used to improve the skin to skin practice in Ethiopia.

6. PLOS authors have the option to publish the peer review history of their article (what does this mean?). If published, this will include your full peer review and any attached files.

Reviewer #1: No

Reviewer #2: No

---

## [Author Response · Author response to Decision Letter 0]

18 Oct 2021

Revision 

Please ensure that your manuscript meets PLOS ONE's style requirements, including those for file naming. (Ensured)

Please amend your current ethics statement to address the following concerns: Please explain why written consent was not obtained, how you recorded/documented participant consent: 

Answer: it is optional to use the written or verbal consent.

For this study verbal consent was used because the participants permit to take only verbal consent and the consent taken by reading the format prepared with the questioner and checklist, i.e. ‘’ the data collectors has been read the informed consent for the participants by informing them about benefit, procedures, duration, alternatives of participation (whether participated or not) and confidentiality. By asking the participants the question ‘’could I have your permission to continue?’’

1. If yes, will continue to distribute the questionnaire. 

2. If no, skip to the next participant by writing reasons for his/her refusal.

3. Please include additional information regarding the checklist and questionnaire used in the study and ensure that you have provided sufficient details that others could replicate the analyses. For instance, if you developed a questionnaire as part of this study and it is not under a copyright more restrictive than CC-BY, please include a copy, in both the original language and English.

Answer: Self-administered questionnaire and observational checklist were developed by reviewing different related literatures, and prepared in English.

The following is the copy of questionnaire and Checklist

Annexes:

ENGLISH VERSION QUESTIONNAIRE AND CHECKLIST

Practice of early mother-newborn skin-to-skin contact after delivery of healthy term neonate and associated factors among health care providers at health facilities of Southwestern Oromia, Ethiopia 2017.

Greeting: 

Hello, My name is_____________________. I am here today to collect data on Assessment of Practice of early mother-newborn skin-to-skin contact (SSC) after delivery of healthy term neonate and associated factors among health care providers at health facilities of Southwestern Oromia, Ethiopia 2017. The purpose of this study is to explore and describe about health care providers’ practice related to early SSC and its associated factors. I request you to take part in this study and to respond genuinely. 

Your cooperation and willingness is greatly helpful in identifying problems related to early SSC in mothers who gave birth. The study will be conducted through self-administer questionnaire and you are being asked for a little of your time, about 20 min, to help us in this study. 

Your name will not be written in this form and will never be used in connection with any information you will tell us. There is no possible risk associated with participating in this study except the time spent for responding to the questionnaire. All information given by you will be kept strictly confidential. Your participation will be voluntary and you are not obligated to answer any question you do not wish to answer. If you feel discomfort with the question, it is your right to drop it any time you want. If you have questions regarding this study or would like to be informed of the results after its completion.

Could I have your permission to continue? 

1. If yes, will continue to distribute the questionnaire. 

2. If no, skip to the next participant by writing reasons for his/her refusal 

Informed consent Certified by 

Data collectors Name--------------------------------signature------------------- 

Date of Data collection-----------------Time started---------------------- Time completed----------

Result of data collection: 

 1. Completed--------- 

 2. Respondent not available--------

 3. Refused------ 

 4. Partially completed......... 

Checked by..............................................................

ANNEX-A: CONSENT FORM AND QUESTIONNAIRE 

Consent form before distributing the q

---

## [Decision Letter · Decision Letter 1]

30 Dec 2021

PONE-D-21-11913R1The Practice of early mother-newborn skin-to-skin contact after delivery of healthy term neonate and associated factors among health care providers at health facilities of Southwestern Ethiopia: A cross-sectional studyPLOS ONE

Dear Dr. Dejene Edosa Dirirsa,

Thank you for submitting your manuscript to PLOS ONE. After careful consideration, we feel that it has merit but does not fully meet PLOS ONE’s publication criteria as it currently stands. Therefore, we invite you to submit a revised version of the manuscript that addresses the points raised during the review process.

Please address MINOR comments by the reviewers 

We look forward to receiving your revised manuscript.

Kind regards,

Sajid Bashir Soofi

Academic Editor

PLOS ONE

Journal Requirements:

Additional Editor Comments (if provided):

Please address point by point some minor comments by the reviewer

Reviewers' comments:

Reviewer's Responses to Questions

**Comments to the Author**

1. If the authors have adequately addressed your comments raised in a previous round of review and you feel that this manuscript is now acceptable for publication, you may indicate that here to bypass the “Comments to the Author” section, enter your conflict of interest statement in the “Confidential to Editor” section, and submit your "Accept" recommendation.

Reviewer #1: (No Response)

Reviewer #2: All comments have been addressed

2. Is the manuscript technically sound, and do the data support the conclusions?

Reviewer #1: (No Response)

Reviewer #2: Yes

3. Has the statistical analysis been performed appropriately and rigorously? 

Reviewer #1: (No Response)

Reviewer #2: Yes

4. Have the authors made all data underlying the findings in their manuscript fully available?

Reviewer #1: (No Response)

Reviewer #2: Yes

5. Is the manuscript presented in an intelligible fashion and written in standard English?

Reviewer #1: (No Response)

Reviewer #2: No

6. Review Comments to the Author

Reviewer #1: The title and objective is not consistent with the results of the article

Items such as maternal factors are mentioned in the results that are not mentioned in the title or objective

The number of references is low, Suggested articles for use in this manuscript

1. Karimi FZ, Sadeghi R, Maleki-Saghooni N, Khadivzadeh T. The effect of mother-infant skin to skin contact on success and duration of first breastfeeding: A systematic review and meta-analysis. Taiwanese Journal of Obstetrics and Gynecology. 2019 Jan 1; 58(1):1-9.

2. Karimi FZ, MIRI HH, Salehian M, Khadivzadeh T, Bakhshi M. The Effect of Mother-Infant Skin to Skin Contact after Birth on Third Stage of Labor: A Systematic Review and Meta-Analysis. Iranian journal of public health. 2019 Apr;48(4):612.

3. karimi FZ, Miri HH, Khadivzadeh T, Maleki-Saghooni N. The effect of mother-infant skin-to-skin contact immediately after birth on exclusive breastfeeding: a systematic review and meta-analysis. Journal of the Turkish German Gynecological Association. 2020 Mar;21(1):46

4. Karimi A, Bagheri S, Khadivzadeh T, Mirzaii Najmabadi Kh. The Effect of an Interventional Program, Based on the Theory of Ethology, on Breastfeeding Competence of Infants. Iranian Journal of Neonatology 2014; 5(3): 10-12.

5. Karimi FZ, Khadivzadeh T, Saeidi M, Bagheri S. The Effect of Kangaroo Mother Care Immediately after Delivery on Mother-infant Attachment and on Maternal Anxiety about the Baby 3- Months after Delivery: a Randomized Controlled Trial. Int J Pediatr 2016; 4(9): 3561-70

6. Karimi A, Tara F, Khadivzadeh T, Aghamohammadian Sharbaf HR. The Effect of Skin to Skin Contact Immediately after Delivery on the Maternal Attachment and Anxiety Regarding Infant. The Iranian Journal of Obstetrics, Gynecology and Infertility 2013; 16(67): 7-15.

7. Karimi FZ, Bagheri S, Tara F, Khadivzadeh T, Mousavi Bazaz SM. Effect of Kangaroo Mother Care on breastfeeding self-efficacy in primiparous women, 3 month after child birth. The Iranian Journal of Obstetrics, Gynecology and Infertility 2014; 17(120): 1-8.

8. Khadivzadeh T, Karimi FZ, Tara F, Bagheri S. The Effect of Postpartum Mother– Infant Skin-to-Skin Contact on Exclusive Breastfeeding In neonatal period: A Randomized Controlled Trial. Int J Pediatr 2016; 4(5): 5409-17.

9. Khadivzadeh, T., Karimi, F., Tara, F. Effects of early mother-neonate skin-to-skin contact on the duration of the third stage of labor: A randomized clinical trial. The Iranian Journal of Obstetrics, Gynecology and Infertility, 2018; 21(2): 23-29

Reviewer #2: There are several gramatical error of the manuscript, especially in the abstract section that may benefit from copyediting. Thank you.

7. PLOS authors have the option to publish the peer review history of their article (what does this mean?). If published, this will include your full peer review and any attached files.

Reviewer #1: No

Reviewer #2: No

---

## [Author Response · Author response to Decision Letter 1]

17 Feb 2022

Tittle: The Practice of early mother-newborn skin-to-skin contact after delivery of healthy term neonate and associated factors among health care providers at health facilities of Southwestern Ethiopia: A cross-sectional study 

The responses of reviewed manuscript for academic editor & reviewers

Academic Editor

Comment 1: Please review your reference list to ensure that it is complete and correct. If you have cited papers that have been retracted, please include the rationale for doing so in the manuscript text, or remove these references and replace them with relevant current references. Any changes to the reference list should be mentioned in the rebuttal letter that accompanies your revised manuscript. If you need to cite a retracted article, indicate the article’s retracted status in the References list and also include a citation and full reference for the retraction notice.

Response: We have used five references that recommended by the reviewer #1 (Ref: Number 2, 6, 7, 8, 12): these are listed below under the response for reviewer #1.

Reviewer#1: 

Comment 1: The title and objective is not consistent with the results of the article

Items such as maternal factors are mentioned in the results that are not mentioned in the title or objective.

Response: thank you for your constructive comments. 

• Un-necessary factors mentioned in the result like ‘maternal factors’ removed from the result part, which are not related with title and objectives.

Comment 2: The number of references is low, Suggested articles for use in this manuscript. 

• I considered the references that you suggested and 5 of them cited (Ref: Number 2, 6, 7, 8, 12). These are:

• Karimi FZ, Sadeghi R, Maleki-Saghooni N, Khadivzadeh T. The effect of mother-infant skin to skin contact on success and duration of first breastfeeding: A systematic review and meta-analysis. Taiwanese Journal of Obstetrics and Gynecology. 2019 Jan 1; 58(1):1-9.

• Karimi A, Bagheri S, Khadivzadeh T, Mirzaii Najmabadi Kh. The Effect of an Interventional Program, Based on the Theory of Ethology, on Breastfeeding Competence of Infants. Iranian Journal of Neonatology 2014; 5(3): 10-12.

• Karimi FZ, Khadivzadeh T, Saeidi M, Bagheri S. The Effect of Kangaroo Mother Care Immediately after Delivery on Mother-infant Attachment and on Maternal Anxiety about the Baby 3- Months after Delivery: a Randomized Controlled Trial. Int J Pediatr 2016; 4(9): 3561-70.

• Karimi FZ, MIRI HH, Salehian M, Khadivzadeh T, Bakhshi M. The Effect of Mother-Infant Skin to Skin Contact after Birth on Third Stage of Labor: A Systematic Review and Meta-Analysis. Iranian journal of public health. 2019 Apr;48(4):612.

• Khadivzadeh T, Karimi FZ, Tara F, Bagheri S. The Effect of Postpartum Mother–Infant Skin-to-Skin Contact on Exclusive Breastfeeding In neonatal period: A Randomized Controlled Trial. Int J Pediatr 2016; 4(5): 5409-17. DOI: 10.22038/ijp.2016.7522. 

Reviewer#2: 

Comment 1: There are several grammatical error of the manuscript, especially in the abstract section that may benefit from copyediting. 

Response: Thank you for your positive and constructive comments. As much as possible I tried to correct the whole document for grammatical errors including the abstract part.

---

## [Decision Letter · Decision Letter 2]

18 Jul 2022

PONE-D-21-11913R2The Practice of early mother-newborn skin-to-skin contact after delivery of healthy term neonate and associated factors among health care professionals at health facilities of Southwestern Oromia, Ethiopia: A cross-sectional studyPLOS ONE

Dear Dr. Dirirsa,

Thank you for submitting your manuscript to PLOS ONE. After careful consideration, we feel that it has merit but does not fully meet PLOS ONE’s publication criteria as it currently stands. Therefore, we invite you to submit a revised version of the manuscript that addresses the points raised during the review process.

Specifically, we require that the article is presented in an intelligible fashion and is written in standard English. We noted multiple language issues, especially in Abstract. Please have your manuscript carefully copyedited and correct any language errors this time In addition, we note that one or more reviewers has recommended that you cite specific previously published works. As always, we recommend that you please review and evaluate the requested works to determine whether they are relevant and should be cited. It is not a requirement to cite these works. For instance, we feel that the suggested references 2,5,6,7,8, and 12 are not so relevant and you are optional to cite them. Furthermore, We note that you have indicated that “All relevant data are within the manuscript”. However, members of the editorial team have assessed the provided data and are concerned that the data provided do not meet our expectations for minimal datasets. PLOS defines the minimal data set to consist of the data required to replicate all study findings reported in the article, as well as related metadata and methods (see (https://journals.plos.org/plosone/s/data-availability). For example, authors should submit the following data:

> The values behind the means, standard deviations and other measures reported;

> The values used to build graphs;

> The points extracted from images for analysis.

Please ensure that you have provided a datafile to meet these requirements with your manuscript.

Finally, we noted the following conflicted data statements:

-On the submission details page: "No - some restrictions will apply" and "All relevant data are within the manuscript"

-In the manuscript: "The data used or analyzed throughout the current study will be obtained up on request from the corresponding author and coauthors." Please comment on this and keep your data statements consistent.

We look forward to receiving your revised manuscript.

Kind regards,

Jianhong Zhou

Staff Editor

PLOS ONE

Journal Requirements:

Reviewers' comments:

Reviewer's Responses to Questions

**Comments to the Author**

1. If the authors have adequately addressed your comments raised in a previous round of review and you feel that this manuscript is now acceptable for publication, you may indicate that here to bypass the “Comments to the Author” section, enter your conflict of interest statement in the “Confidential to Editor” section, and submit your "Accept" recommendation.

Reviewer #1: All comments have been addressed

Reviewer #2: All comments have been addressed

2. Is the manuscript technically sound, and do the data support the conclusions?

Reviewer #1: Yes

Reviewer #2: Yes

3. Has the statistical analysis been performed appropriately and rigorously? 

Reviewer #1: Yes

Reviewer #2: Yes

4. Have the authors made all data underlying the findings in their manuscript fully available?

Reviewer #1: Yes

Reviewer #2: Yes

5. Is the manuscript presented in an intelligible fashion and written in standard English?

Reviewer #1: Yes

Reviewer #2: Yes

6. Review Comments to the Author

Reviewer #1: Please use the space provided to explain your answers to the questions above. You may also include additional comments for the author, including concerns about dual publication, research ethics, or publication ethics. (Please upload your review as an attachment if it exceeds 20,000 characters)

Reviewer #2: Dear Author, thank you for preparing the corrections. Improvements have been made to the manuscript and is satisfactory. All comments of the reviewers were answered. Well done!

7. PLOS authors have the option to publish the peer review history of their article (what does this mean?). If published, this will include your full peer review and any attached files.

Reviewer #1: No

Reviewer #2: **Yes: **Assoc. Prof Dr F Ariffin

---

## [Author Response · Author response to Decision Letter 2]

28 Aug 2022

Tittle: The Practice of early mother-newborn skin-to-skin contact after delivery of healthy term neonate and associated factors among health care providers at health facilities of Southwestern Ethiopia: A cross-sectional study 

The responses of reviewed manuscript for academic editor & reviewers

Academic Editor

Comment 1: We noted multiple language issues, especially in Abstract. Please have your manuscript carefully copyedited and correct any language errors this time

Response: We have tried to correct the grammatical errors in th abstract section of the manuscript. 

Comment 2: In addition, we note that one or more reviewers has recommended that you cite specific previously published works. As always, we recommend that you please review and evaluate the requested works to determine whether they are relevant and should be cited. It is not a requirement to cite these works. For instance, we feel that the suggested references 2,5,6,7,8, and 12 are not so relevant and you are optional to cite them.

Response: the citations that said irrelevant to our study are removed according to your suggestion. 

Comment 3: Furthermore, We note that you have indicated that “All relevant data are within the manuscript”. Finally, we noted the following conflicted data statements:

-On the submission details page: "No - some restrictions will apply" and "All relevant data are within the manuscript" -In the manuscript: "The data used or analyzed throughout the current study will be obtained up on request from the corresponding author and coauthors." Please comment on this and keep your data statements consistent.

Response: we make data statements consistent according to your comment. 

Reviewer #1: ‘’Please use the space provided to explain your answers to the questions above. You may also include additional comments for the author, including concerns about dual publication, research ethics, or publication ethics. (Please upload your review as an attachment if it exceeds 20,000 characters)’’

Reviewer #2: ‘’Dear Author, thank you for preparing the corrections. Improvements have been made to the manuscript and are satisfactory. All comments of the reviewers were answered. Well done!’’

Thank you the reviewers and editors for your constructive comments and suggestions. We gain so many experiences from your valuable comments and suggestions. Thank you again!

---

## [Editor Report · Decision Letter 3]

1 Sep 2022

The Practice of early mother-newborn skin-to-skin contact after delivery of healthy term neonate and associated factors among health care professionals at health facilities of Southwestern Oromia, Ethiopia: A cross-sectional study

PONE-D-21-11913R3

Dear Dr. Dirirsa,

We’re pleased to inform you that your manuscript has been judged scientifically suitable for publication and will be formally accepted for publication once it meets all outstanding technical requirements.

Kind regards,

Jianhong Zhou

Staff Editor

PLOS ONE

Additional Editor Comments : We still noted multiple language issues. Please pay attention to the Introduction section and have your whole manuscript carefully copyedited and correct any language errors this time.
---

## [Editor Report · Acceptance letter]

5 Dec 2022

PONE-D-21-11913R3 

The Practice of early mother-newborn skin-to-skin contact after delivery of healthy term neonate and associated factors among health care professionals at health facilities of Southwestern Oromia, Ethiopia: A cross-sectional study 

Dear Dr. Dirirsa:

I'm pleased to inform you that your manuscript has been deemed suitable for publication in PLOS ONE. Congratulations! Your manuscript is now with our production department. 

Kind regards, 

on behalf of

Jianhong Zhou 

Staff Editor

PLOS ONE